# CRISPRi screen of long non-coding RNAs identifies *LINC03045* regulating glioblastoma invasion

Kathleen Tsung[1], Kristie Q. Liu[1]*, Jane S. Han[1], Krutika Deshpande[1],
Tammy Doan[2], Yong-Hwee Eddie Loh[3], Li Ding[4], Wentao Yang[2], Josh Neman[5],
Yali Dou[2,6], Frank J. Attenello[1,2]

1 Department of Neurological Surgery, Keck School of Medicine, University of Southern California, Los
Angeles, California, United States of America, 2 Department of Biochemistry and Molecular Medicine, Keck
School of Medicine, University of Southern California, Los Angeles, California, United States of America,
3 USC Libraries Bioinformatics Services, University of Southern California, Los Angeles, California, United
States of America, 4 Department of Preventative Medicine, Keck School of Medicine, University of Southern
California, Los Angeles, California, United States of America, 5 Department of Neurological Surgery,
Physiology and Neuroscience, USC Brain Tumor Center, Norris Comprehensive Cancer Center, Keck School
of Medicine, University of Southern California, Los Angeles, California, United States of America,
6 Department of Medicine, Keck School of Medicine, University of Southern California, Los Angeles,
California, United States of America

* kqliu@usc.edu

## Abstract

### Introduction

Glioblastoma (GBM) invasion studies have focused on coding genes, while few studies
evaluate long non-coding RNAs (lncRNAs), transcripts without protein-coding potential, for
role in GBM invasion. We leveraged CRISPR-interference (CRISPRi) to evaluate invasive
function of GBM-associated lncRNAs in an unbiased functional screen, characterizing and
exploring the mechanism of identified candidates.

### Methods

We implemented a CRISPRi lncRNA loss-of-function screen evaluating association of
lncRNA knockdown (KD) with invasion capacity in Matrigel. Top screen candidates were
validated using CRISPRi and oligonucleotide(ASO)-mediated knockdown in three tumor
lines. Clinical relevance of candidates was assessed via The Cancer Genome Atlas(TCGA)
and Genotype-Tissue Expression(GTEx) survival analysis. Mediators of lncRNA effect were
identified via differential expression analysis following lncRNA KD and assessed for tumor
invasion using knockdown and rescue experiments.

### Results

Forty-eight lncRNAs were significantly associated with 33–83% decrease in invasion
(p<0.01) upon knockdown. The top candidate, *LINC03045*, identified from effect size and p-
value, demonstrated 82.7% decrease in tumor cell invasion upon knockdown, while
*LINC03045* expression was significantly associated with patient survival and tumor grade

pgen.1011314

STATES

**Data Availability Statement:** The data that support
the findings of this study are openly available in the
NCBI Gene Expression Omnibus (GEO) database,
accession number GSE268342. Other data utilized

in this study are available publicly in the Genotype-Tissue Expression Project, Gene Ontology, The Cancer Genome Atlas, and the Kyoto Encyclopedia of Genes and Genomes databases.

**Funding:** FJA was funded by the Margaret E. Early Medical Research Trust (https://rii.usc.edu/limited-submissions/margaret-e-early/), NIH K08 Grant 5K08NS114172-02 (https://www.nia.nih.gov/research/training/k08-mentored-clinical-scientist-development-awards), NIH KL2 Grant 5KL2TR001854-04, and the Keck School of Medicine Dean's Pilot Funding Program (https://keck.usc.edu/research-funding/programs-administered-by-ksom/). LD was supported by Grant UL1TR001855 from the National Center for Advancing Translational Science (NCATS) of the U.S. National Institutes of Health (https://ncats.nih.gov/). The funders had no role in study design, data collection and analysis, decision to publish, or preparation of the manuscript. The authors did not receive a salary from any of the funders.

**Competing interests:** The authors have declared that no competing interests exist.

(p<0.0001). RNAseq analysis of *LINC03045* knockdown revealed that *WASF3*, previously implicated in tumor invasion studies, was highly correlated with lncRNA expression, while *WASF3* KD was associated with significant decrease in invasion. Finally, *WASF3* overexpression demonstrated rescue of invasive function lost with *LINC03045* KD.

## Conclusion

CRISPRi screening identified *LINC03045*, a previously unannotated lncRNA, as critical to GBM invasion. Gene expression is significantly associated with tumor grade and survival. RNA-seq and mechanistic studies suggest that this novel lncRNA may regulate invasion via *WASF3*.

## Author summary

Glioblastoma is the most common malignant brain tumor to originate from brain tissue in adults. Most research has focused on the role of genes that code for protein, but genes that do not code for protein can still produce long non-coding RNAs (lncRNAs) that have been demonstrated to have important functions in cancer processes. We utilized a method called CRISPRi to screen for lncRNAs that are important in glioblastoma invasion, which is the capacity of cancer cells to invade into surrounding tissue. With this CRISPRi invasion screen, we identified *LINC03045* as a gene that affects glioblastoma invasion. We validated this finding using cultured cancer stem cells and found that knocking down this gene decreased invasion. We also analyzed the public databases for clinical relevance, and found that *LINC03045* was associated with patient survival and tumor grade. We also identified the gene *WASF3*, previously implicated in multiple tumor invasion studies (an element of the JAK-STAT pathway) as a downstream element of *LINC03045*. Thus, here, we have identified *LINC03045* as a critical non-coding gene in glioblastoma invasion, and further characterizing this gene could potentially make it a therapeutic target for glioblastoma therapy in the future.

## Introduction

Glioblastoma (GBM) is the most common and aggressive adult primary malignant brain tumor, with an estimated annual incidence of 17,000 new cases per year in the USA [1]. Standard of care treatment includes surgical resection, chemotherapy with temozolomide (TMZ), and radiation therapy, but despite the available therapeutic options, mean survival remains at about 12 to 15 months and the five-year survival rate at 5.5% [2–5]. The majority of GBM patients also invariably experience tumor recurrence with increased chemo- and radio-resistance, and there is no standardized second line of management after initial adjuvant treatment. Further evidence suggests that microscopic tumor invasion beyond gross radiographic and intraoperatively visualized tumor specimen contributes to continued tumor growth despite gross total radiographic resection [6–8]. The limited efficacy of treatment options necessitates novel therapeutic strategies and molecular targets that can limit this tumor invasion and subsequent tumor recurrence.

While current genetic studies in glioma research have primarily focused on targeting coding genes, about 80% of the genome is composed of non-coding RNAs [9,10]. Long non-

coding RNAs (lncRNAs) are transcripts greater than 200 nucleotides that do not code for proteins but have demonstrated involvement in transcriptional and post-transcriptional gene regulation. LncRNAs have been shown to play roles in the pathogenesis of multiple malignancies. Notably, because they exhibit highly cell type-specific expression and function, they have become attractive novel targets of gene therapy [9,11–18]. To identify novel genetic therapeutic targets in cancer treatment, systematic functional screens have recently been successfully employed to rapidly and efficiently identify transcripts could sensitize cancer cells to treatment. Despite the rapidly emerging use of these functional screens to identify novel genetic targets, few studies have specifically identified novel candidate lncRNAs targeting glioma.

Clustered regularly interspaced short palindromic repeats (CRISPR) is a technology that has allowed for genome-wide screens of gene function [19–26]. This technology has been valuable in the identification of both coding and non-coding genes that are involved in various cellular phenotypes. Due to the nature of non-coding RNAs, standard CRISPR knockdown may not effectively limit lncRNA function, as small deletions do not typically reduce lncRNA-induced effects [20,23,27]. CRISPRi (CRISPR interference) circumvents this challenge, repressing transcription of the lncRNA at the nucleus via fusion of a catalytically inactive Cas9 (dCas9) with a repressive KRAB domain [28]. We have previously successfully utilized these CRISPRi screens to assess *in vitro* and *in vivo* GBM proliferation [19,29]. CRISPRi screens simultaneously repress multiple genetic targets, with each cell receiving specific individual lncRNA repression. These pools of lncRNA cellular perturbation are then selected for those cells displaying a phenotype of interest. Association of the specific lncRNAs knocked down in remaining selected cells then identifies the lncRNAs that are most closely associated with the selected phenotype.

Here, we developed a CRISPRi-based invasion screen to identify lncRNAs that affect the *in vitro* invasive capacity of glioma cells. In this screen of 2,307 candidate lncRNAs, we sought to identify specific lncRNAs associated with glioma cell invasion. We further assess the clinical association of candidate lncRNAs with tumor grade and patient outcomes while also evaluating potential mechanisms by finding associations of candidate lncRNAs with potential coding genes affecting GBM invasion.

## Results

### CRISPRi screen identifies lncRNAs that affect invasion

To screen for and identify lncRNAs affecting *in vitro* glioma cell invasion, we utilized an invasion screen with CRISPRi as a tool for gene knockdown. Repression of transcription occurred when CRISPRi recruited the Cas9 protein fused to a KRAB repressor (dCas9-KRAB), which is guided and targeted to transcriptional start sites (TSS) by single guide RNAs (sgRNAs) [19]. A GBM cell line (U87) was successfully tranduced with a custom plasmid to express dCas9-KRAB (U87-dCas9-KRAB) fused to blue fluorescent protein (U87-dCas9-KRAB-BFP), and sorted by FACS to a pure population of U87-dCas9-KRAB. Transduced cells were successfullly infected with lentivirus that contained a U87-specific CRISPRi library derived from CRinCL (Addgene #86542, a gift from Jonathan Weissman). This library was comprised of sgRNAs targeting 2,307 lncRNA, with 10 sgRNAs per lncRNA, and 247 negative control sgRNAs, for a total of 23,564 sgRNAs. U87 cells stably expressing dCas9-KRAB were infected such that there was 1,000X coverage per sgRNA, and we selected for infected cells using puromycin (Fig 1A). A low infection efficiency (MOI < 0.3) was utilized to decrease chances of cells being infected by more than one sgRNA. Cells were then seeded in Matrigel-coated Boyden chambers for 24 hours at a density of $5 \times 10^5$ cells per chamber, with serum-free medium in the top chamber and 10% fetal bovine serum medium in the bottom chamber as the chemoattractant for 24

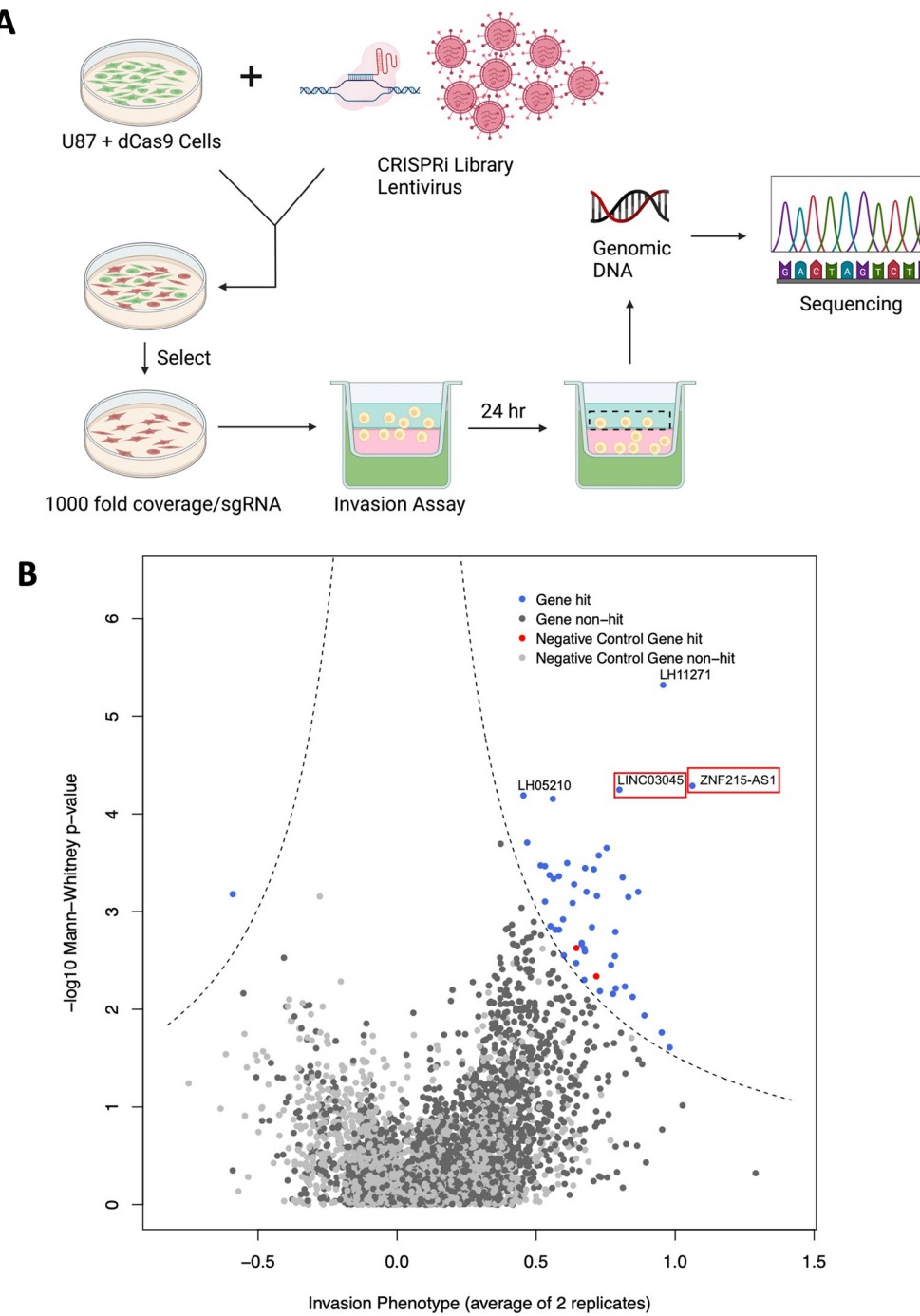

**Fig 1. CRISPRi screen protocol results. A)** Schematic of the screening protocol. The CRISPRi library was packaged into lentivirus and transduced into U87 cells stably expressing dCas9-KRAB, selected for to obtain 1000 fold coverage/sgRNA, and seeded onto Matrigel Boyden chambers. After 24 hours, cells in the top chamber were harvested and processed for deep sequencing. Created by Biorender.com. **B)** Volcano plot of invasion phenotype and negative $\log_{10}$(Mann-Whitney p-value). Screen replicates were averaged and the top sgRNAs for each lncRNA compared to non-targeting controls were used to determine screen hits. The dashed lines represent thresholds to determine screen hits.

hours. At the completion of this invasion period, screen population cells remaining in the top chamber were successfully isolated and collected. DNA was extracted from cells, followed by PCR amplification and isolation of lentivirally incorporated sgRNA screen fragments, prior to analysis by deep sequencing.

The process was repeated to attain two independent replicates, which were averaged together for downstream analysis (S2 Fig). To determine the lncRNAs with the most notable effects on invasion, screen scores, which were defined as the average phenotype of the top three sgRNAs for a given gene multiplied by the negative $\log_{10}$(Mann-Whitney p-value), were compared to non-targeting controls and plotted on a volcano plot, with the dashed lines representing the threshold to determine screen hits (Figs 1B and S1). These dashed lines were chosen based on threshold cutoffs of prior screens, with the threshold for the y-axis cutoff was -log (0.05). This chosen threshold selects lncRNAs with a minimum effect size of greater than approximately 25% at the p-values approach zero (at the highest y-axis -log[p-value] points). Our screen score thus incorporates both p-value and effect size, as previously described [19]. In the non-invading cells, 45 gene targets were significantly enriched in the library, which represents genes that are necessary for glioma cell invasion. Positive hits were identified through this process, and the top candidate was identified as *LINC03045* (annotation LH02236) on chromosome 10. Another candidate, *ZNF215-AS1* (annotation LH02727), was also identified on chromosome 11, but this study will focus on *LINC03045* since subsequent investigation demonstrated that it had a stronger phenotype. Preliminary results, including clinical correlations with tumor grade, patient survival, and effects of *in vitro* knockdown on invasion and growth for *ZNF215-AS1* are shown in S3 Fig.

## Patient outcome analysis via TCGA and GTEx analysis

Since *LINC03045* was previously unannotated in genome maps, we utilized position coordinates to assess expression levels of *LINC03045*, measured in counts per million (CPM). Both GBM (3.6±0.27 CPM) and LGG (2.06±0.30 CPM) showed higher average *LINC03045* gene expression than normal brain cortex (0.67±0.06 CPM) (mean ± SEM) (Fig 2A). Pairwise comparisons of *LINC03045* in tumors vs control showed a significant fold change in expression (GBM vs control: 6.70, p≤0.001; LGG vs control: 6.25, p≤0.001).

Kaplan-Meier (KM) survival analysis of patients with all gliomas (stages 2–4), was conducted based on *LINC03045* counts. The median expression level was 1.65 CPM (IQR 0.72–3.09) and the median survival time was 460.5 days (IQR 231.3–854.3). Log-rank tests for this cohort showed that patient cohorts with elevated expression of *LINC03045* were significantly associated with decreased survival relative to patient cohorts with lower *LINC03045* expression (p<0.0001) (Fig 2B, left). A Cox regression analysis of this cohort showed a dose-response between *LINC03045* expression and survival when the expression level was < 2.5 CPM. The risk maximized and remained constant when the expression level was > = 2.5 CPM. The hazard ratio (HR) is 1.99 (95% CI: 1.61, 2.45, p<0.0001) per 1 CPM increase when the expression level is 0–2.5 CPM. HR is 0.99 (95% CI 0.95–1.03, p = 0.61) when the expression level is > = 2.5 CPM (Fig 2B, right).

We also analyzed survival adjusting both for glioma grade and IDH status. In the GBM cohort (stage 4), there was no significant difference in survival with changes in *LINC03045* expression (p = 0.40) (Fig 2C, left). The median expression was 2.77 CPM (IQR 1.46–4.65) median survival was 360 days (IQR 160–532) for the GBM cohort. Because the TCGA stratifies lower grade gliomas (grade 2 and 3) together due to nonspecific grading data on 20 tumors that could be either grade 2 or 3, we conducted an IDH-adjusted survival analysis of this cohort. In the LGG cohort (grades 2–3) including the 20 patients with uncertain grade 2 or 3 tumors, elevated expression of *LINC03045* was significantly associated with a decrease in

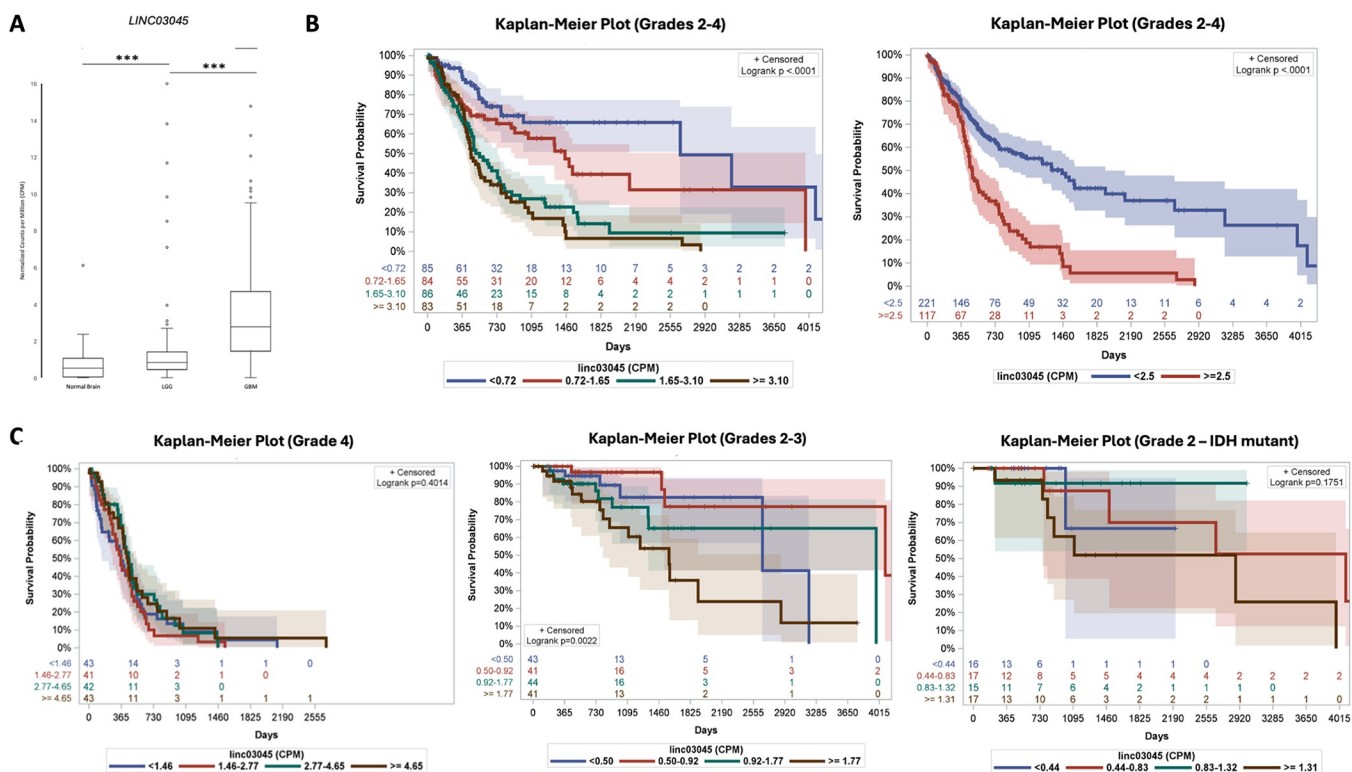

**Fig 2. Expression of *LINC03045* in patient glioma samples from The Cancer Genome Atlas (TCGA) and The Genotype-Tissue Expression (GTEx) project. A)** Boxplots of pairwise comparisons of *LINC03045* expression in normal human brain cortex to low grade glioma (LGG) and glioblastoma (GBM) patient samples. Pairwise comparisons of *LINC03045* expression in patient samples (GBM and LGG) and normal human brain cortex show significantly higher expression in patient tissue than normal tissue (p≤0.001). Expression levels are normalized to standardized genes on the same chromosome (chr10). **B)** Kaplan-Meier survival curves of *LINC03045* for all gliomas (stages 2–4) with expression quantified by normalized counts per million (CPM). Cohorts divided into quartiles (left) show significant decrease in survival with increased expression levels (p<0.0001). Cohorts divided into high vs low risk groups (right) are defined at 2.5 *LINC03045* CPM. The high-risk group (>2.5 CPM) has significantly lower survival (p<0001). Per 1 CPM increase, hazards regression (HR) is 1.99 from 0 to 2.5 CPM. HR meets Cox regression assumption. **C)** Kaplan-Meier survival curves of *LINC03045* for GBM (stage 4, left) and LGG (stages 2–3, middle) adjusted for IDH-status with expression quantified by normalized counts per million (CPM). Cohorts divided into quartiles show a significant decrease in survival with increased expression levels in LGG (p = 0.002), but not GBM (p = 0.40). Survival analysis of IDH-mutant grade 2 alone does not show a significant decrease in survival with increased expression levels (p = 0.18) (right). Position coordinates were used to identify *LINC03045* in RNA-seq data. Expression was normalized to standardized genes on the same chromosome (chr 10). Shading demonstrates the confidence interval. Censored data indicates patients lost to follow-up. *p ≤ 0.05, **p ≤ 0.01, ***p ≤ 0.001.

survival (p = 0.002) (Fig 2C, middle). The median expression was 0.92 CPM (IQR 0.50–1.77) and median survival time was 656 days IQR (417–1294) for this cohort. Notably, there was no significant difference in *LINC0304* expression beetween grade 2 and 3 gliomas within this cohort. A separate survival analysis of IDH-mutant grade 2 gliomas alone (excluding the 20 samples with nonspecific grading) showed that *LINC03045* expression was not associated with survival (p = 0.18) (Fig 2, right). Notably, all analyses were also adjusted for IDH-status, MGMT methylation and 1p/19q codeletion when data was available, though inclusion of variables did not significantly alter effect sizes or p-values of *LINC03045* in Cox regression (not changing from significant to non-significant or vice versa).

## Screen-identified lncRNA phenotypes are reproducible with individual sgRNA knockdown

To validate that the top screen candidates had reproducible phenotypes upon individual lncRNA repression, CRISPRi knockdown (KD) of these gene hits was conducted by

transducing U87 Cas9 cells with the top two sgRNAs for each candidate. Effect size and reproducibility were most apparent with *LINC03045* and the following primary results will focus on *LINC03045*.

Transcriptional expression was evaluated after *LINC03045* CRISPRi KD with qPCR, demonstrating that U87 CRISPRi KD cells had decreased transcriptional expression of *LINC03045* (sgRNA1: 33.4% decrease, p-value = 0.04; sgRNA2: 44.1% decrease, p = 0.02) compared to control (Fig 3A). To validate changes in invasion following CRISPRi KD, invasion assays with *LINC03045* CRISPRi KD were conducted. CRISPRi KD cells demonstrated decreased invasion (sgRNA1: 82.7% decrease, p < 0.0001; sgRNA2: 79.7%, p < 0.0001) (Fig 3C). To ensure that the decreased cell density in Matrigel chambers resulted from chamber invasion, rather than cell death, MTT assays were conducted following *LINC03045* lncRNA KD and demonstrated no change in proliferation in *LINC03045* KD relative to control sgRNA (sgRNA1: p = 0.5, sgRNA2: p = 0.4) (Fig 3B).

## Antisense oligonucleotide mediated KD of *LINC03045* in multiple cell lines replicates CRISPRi results

To assess a whether a second KD method for *LINC03045* is associated with decreased malignant glioma cell invasion and to utilize a drug-based method with translational potential, antisense oligonucleotides (ASOs) were utilized for *LINC03045* KD in three glioma cell lines: a patient-derived tumor line, USC02, and two commercial lines, U87 and U251. Transcriptional expression via qPCR after ASO KD showed decreased transcriptional expression of *LINC03045* in USC02s (33.4% decrease, p = 0.04), U87s (43.9% decrease, p = 0.04), and U251s (51.6% decrease, p = 0.02) (Fig 4A). Likewise, evaluation of *in vitro* invasion via Matrigel assays after ASO-mediated *LINC03045* KD showed decreased invasion in USC02s (44.9% decrease, p = 0.005, Fig 4C), U87s (45.8% decrease, p<0.001, Fig 4D), and U251s (58.4% decrease, p = 0.0017, Fig 4E). Correspondingly, 3D invasion assays in U87 spheroids showed reduced invasion into the surrounding matrix after ASO-mediated *LINC03045* KD, compared to control untransfected U87 spheroids (ASO 1: 34.2% decrease, p = 0.0193, ASO 2: 37% decrease, p = 0.0158, S4C Fig). MTT assays showed no change in cell proliferation after KD of *LINC03045* in USC02s (p = 0.73), U87s (p = 0.15), and U251s (p = 0.51) (Fig 4B).

## Genome-wide differential gene expression via TCGA and GTEx analysis

TCGA and GTEx patient data was further explored for the association of *LINC03045* with genome-wide gene expression changes. The distribution of *LINC03045* expression was first evaluated, noting a clear distinction between the highest and lowest 25 percent of patients (Fig 5A). Subsequently, patients in TCGA-GBM were divided into two groups (high *LINC03045* and low *LINC03045*) to investigate the functional implications of the *LINC03045* based on its expression value. Evaluation of the cohort of patients with low expression of *LINC03045* revealed significant global alterations in gene expression, as represented in the top 100 and 500 genes most altered in association with low lncRNA expression (Fig 5B and 5C). Widespread up- and down-regulation of genes was seen during upregulation of *LINC03045* among patient cohorts as well (Fig 5D). KEGG pathway analysis revealed a significant association between *LINC03045* expression and oxidative phosphorylation, as well as ribosomal and spliceosomal activity (Fig 5E). Gene ontology analysis revealed widespread association of RNA transcription, RNA polymerase II activity, RNA splicing, and chromatin organization with increased *LINC03045* expression (Fig 5F). Among implicated gene ontology pathways, *WASF3* was significantly associated with an enriched pathway involving cytoskeleton modulation (GO:0007010). Further, we evaluated the intersection of differential gene expression profile

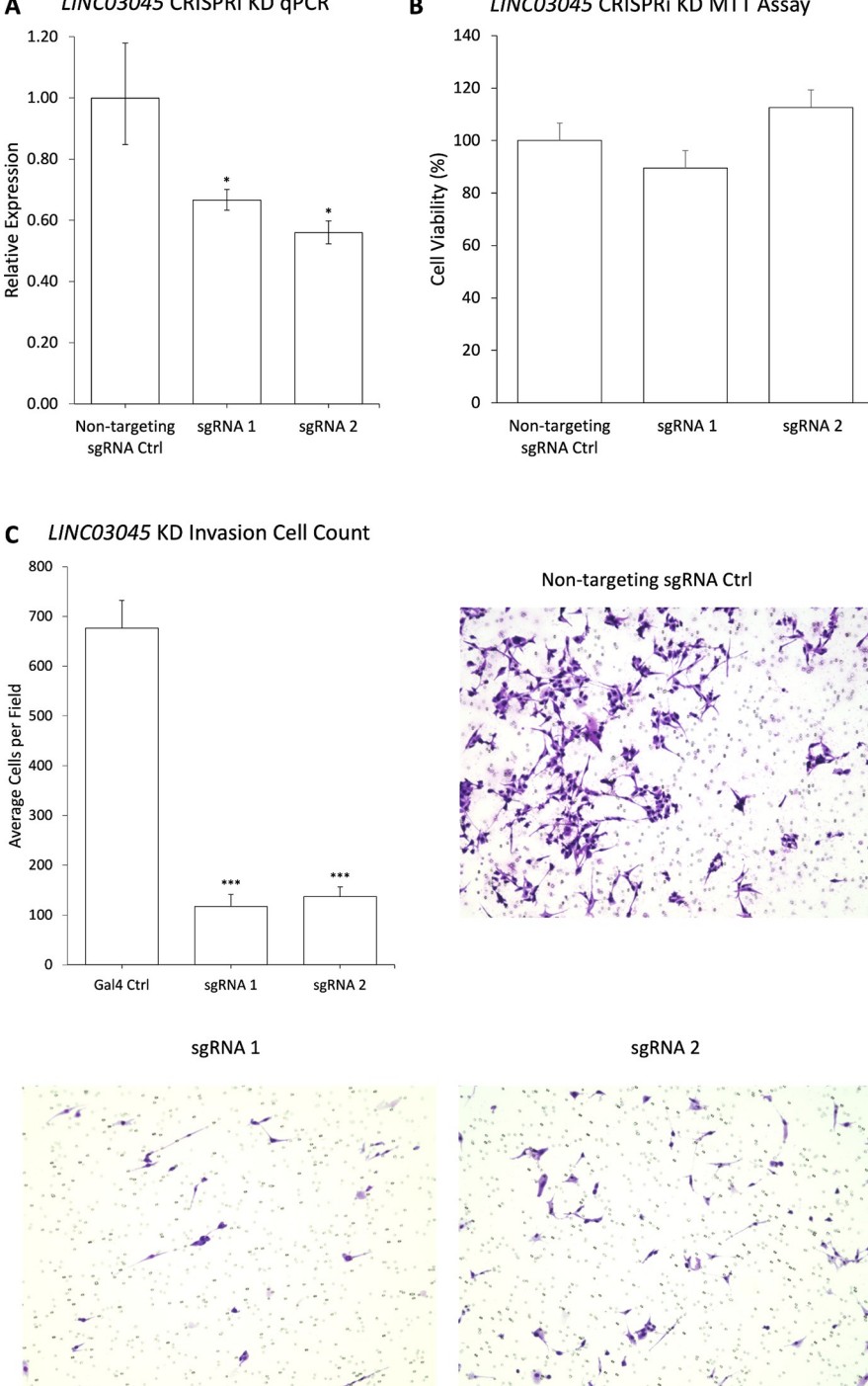

**Fig 3. Validation of CRISPRi screen results. A)** qPCR of CRISPRi knockdown (KD) of the top two sgRNAs for *LINC03045* in U87 cells. All expression levels were normalized to the housekeeping gene *RPLP0*. KD cells were compared to the non-targeting sgRNA control. Data are expressed as mean ± SD for replicates. $*p \leq 0.05$, $**p \leq 0.01$, $***p \leq 0.001$ **B)** An MTT assay showing no change in proliferation after KD of *LINC03045* by CRISPRi in U87 cells. KD cells were compared to the non-targeting sgRNA control. Data are expressed as mean ± SD. $*p \leq 0.05$, $**p \leq 0.01$, $***p \leq 0.001$. **C)** Representative images of invasion assay through a Matrigel-coated Boyden chamber after KD of *LINC03045* by CRISPRi in U87 cells. The bar graph represents the number of invaded cells per field counted, with KD cells compared to the non-targeting sgRNA control. Data are expressed as mean ± SEM of three independent experiments with 9 fields imaged per experiment. $*p \leq 0.05$, $**p \leq 0.01$, $***p \leq 0.001$.

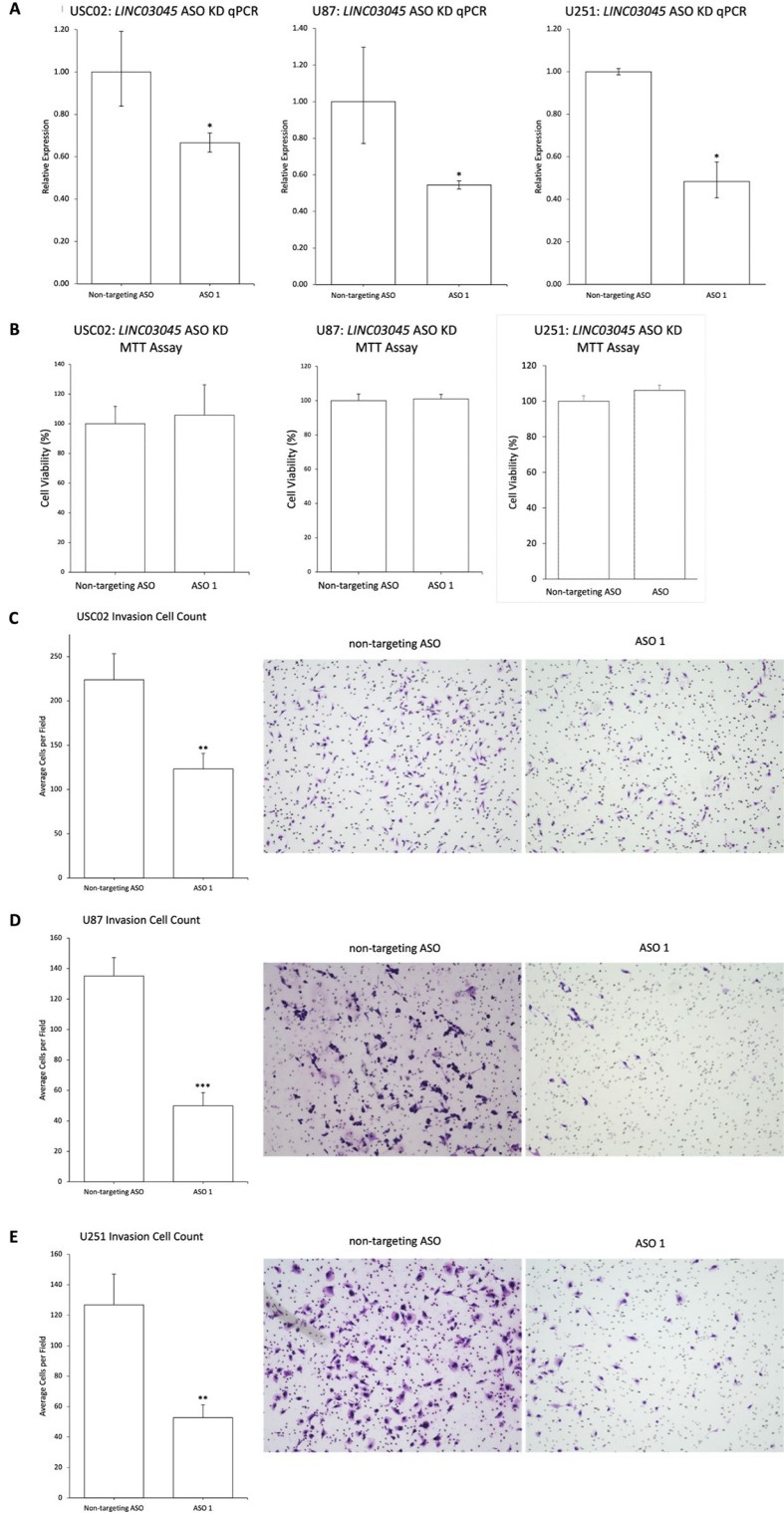

**Fig 4. Antisense oligonucleotide mediated KD of *LINC03045* in multiple cell lines replicates CRISPRi results. A)** qPCR of antisense oligonucleotide (ASO) KD of *LINC03045* in USC02, U87, and U251 cells. All expression levels were normalized to the housekeeping gene *RPLP0*. KD cells were compared to the non-targeting ASO control. Data are expressed as mean ± SD for replicates. *p ≤ 0.05, **p ≤ 0.01, ***p ≤ 0.001 **B)** An MTT assay showing no change in proliferation after KD of *LINC03045* by ASO in USC02, U87, and U251 cells. KD cells were compared to the non-targeting ASO control. Data are expressed as mean ± SD. *p ≤ 0.05, **p ≤ 0.01, ***p ≤ 0.001 **C—E)** Representative

images of invasion assay through a Matrigel-coated Boyden chamber after KD of *LINC03045* in USC02 (C), U87 (D), and U251 (E) cells by ASO. The bar graph represents the number of invaded cells per field counted, with KD cells compared to the non-targeting ASO control. Data are expressed as mean ± SEM of three independent experiments with 9 fields imaged per experiment. *p ≤ 0.05, **p ≤ 0.01, ***p ≤ 0.001.

associated with *LINC03045* KD *in vitro* and *LINC03045* expression changes *in vivo* (Fig 5G). For the TCGA-associated genes, we specifically included protein-coding genes that showed a significant association with *LINC03045* (p< = 0.05). Of 36 differentially expressed genes following *in vitro* linc03045 KD, 16 genes were differentially expressed in the TCGA. Notably, the overlapping genes (e.g. *WASF3*, *EDNRB*, *SOX10*, *BAMBI*) were found to be involved in several pathways, such as regulation of cell shape and neural crest cell migration (Fig 5 and S1 Table).

### WASF3 is a downstream mediator of *LINC03045*

To investigate downstream effects of *LINC03045* and begin elucidating potential mechanisms mediating the *LINC03045* activity, we conducted ASO-mediated KD of *LINC03045* in U87 cells, evaluating gene expression via RNAseq. Results from RNAseq analysis of cells with *LINC03045* knockdown relative to control are shown in the form of a heat map in Fig 6A and a volcano plot in Fig 6B. Several genes were found to be significantly up or downregulated with *LINC03045* knockdown. To further improve the reliability and comparability of detected differential expressed genes, only 4 protein coding genes (*ABCC1*, *DCT*, *RPS17*, *TBCCD1*, and *WASF3*) with RPKM values ≥1 in either control or treatment group were identified. These genes were individually validated for gene expression via qPCR following *LINC03045* knockdown. In addition, when candidate genes were compared to genes associated with *LINC03045* expression in TCGA patient data, we found that *WASF3*, a gene that has previously been found to be associated with the JAK-STAT pathway, was significantly positively correlated with the expression of *LINC03045* (Fig 6C and S1 Table). Finally, the candidates were evaluated for individual knockdown/overexpression and assessment of *in vitro* invasion via Boyden chamber assay.

*WASF3* was uniquely noted to be reproducibly decreased upon *LINC03045* KD, as well as associated with decreased tumor cell invasion upon *WASF3* KD. Furthermore, *WASF3* was the only one of these gene candidates noted to be significantly downregulated among patient cohorts with low *LINC03045* expression in our analysis above (Fig 6B).

Specifically, upon ASO-mediated *LINC03045* KD, we noted a decrease in transcriptional expression of *WASF3* in USC02s (25.9% decrease, p = 0.03), U87s (42.7% decrease, p = 0.002), and U251s (57.3% decrease, p = 0.01) (Fig 7A). Furthermore, upon siRNA-mediated KD of *WASF3*, qPCR confirmed a decrease in *WASF3* transcriptional expression in all USC02s (siRNA1: 47.9% decrease, p = 0.004; siRNA2: 87.3% decrease, p = 0.03), U87s (siRNA1: 64.2% decrease, p = 0.002; siRNA2: 85.9% decrease, p = 0.05), and U251s (siRNA1: 74.7% decrease, p = 0.02; siRNA2: 94.2% decrease, p = 0.007) (Fig 7B). In addition, *WASF3* KD followed by assessment of cell invasion via Matrigel assays demonstrated a statistically significant decrease in invasion relative to control in USC02s (25.7% decrease, p<0.001, Fig 7C), U87s (16.4% decrease, p<0.001, Fig 7D), and U251s (27.3% decrease, p<0.001, Fig 7E). Additionally, 3D invasion assays demonstrated reduced invasion of U87 spheroids into the surrounding matrix after siRNA-mediated *WASF3* KD, compared to control untransfected U87 spheroids (siRNA1: 41.1% decrease, p = 0.0113; siRNA2: 49.4% decrease, p = 0.0071), S4D Fig), Further analysis demonstrated that *WASF3* has a statistically significant positive correlation with *LINC03045* (p = 0.016) (Fig 6C).

To further reinforce the relationship between *LINC03045* and *WASF3*, we also investigated if *WASF3* overexpression could rescue invasive capacity after *LINC03045* KD in the patient-

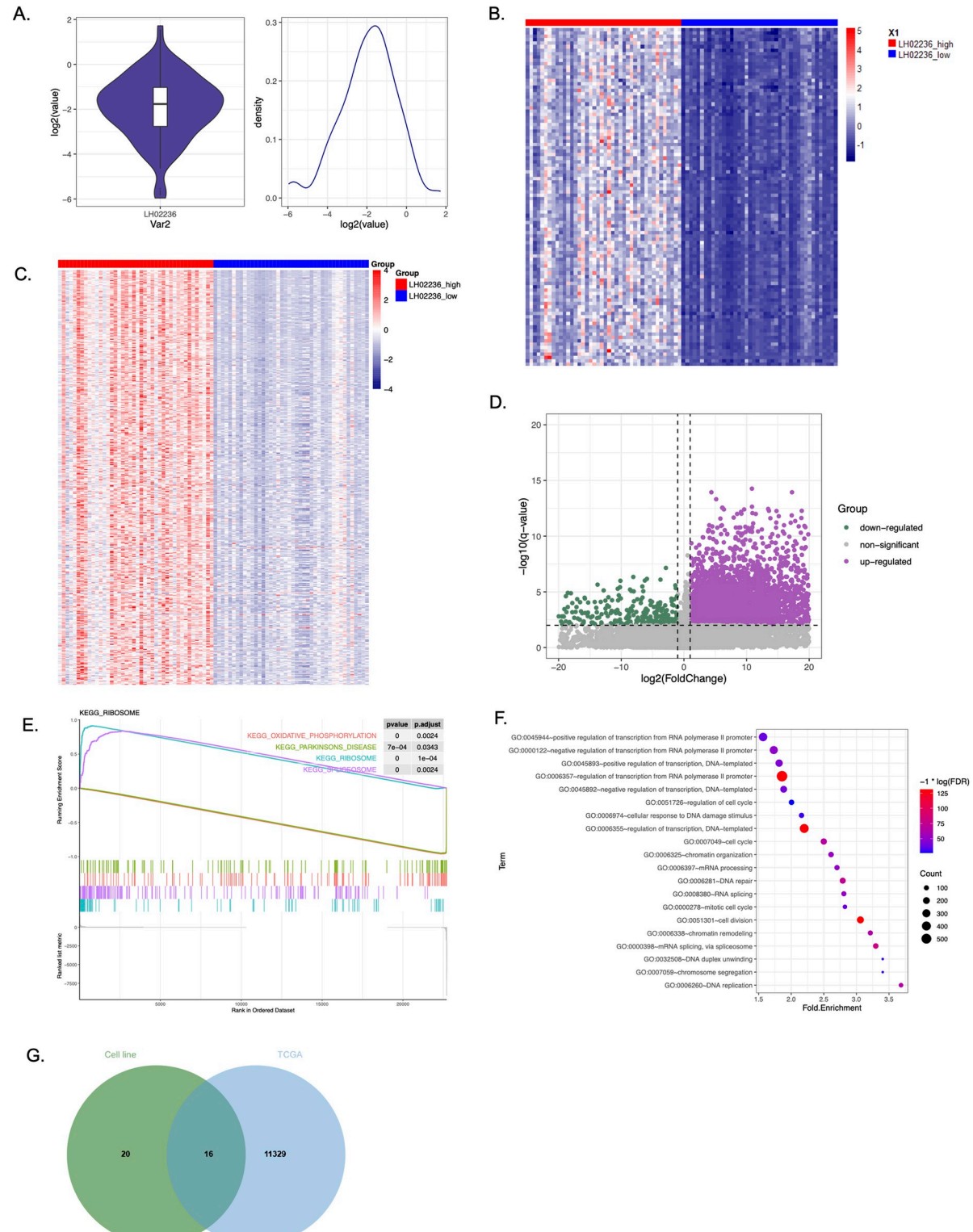

**Fig 5. Genome-wide differential gene expression from The Cancer Genome Atlas (TCGA) and The Genotype-Tissue Expression (GTEx) project. A)** Distribution of *LINC03045* expression utilized to determine high and low *LINC03045* expression groups. **B)** Heat map showing differential expression of the top 100 genes in high vs. low *LINC03045* expression groups. **C)** Heat map showing differential expression of the top 500 genes in high vs. low *LINC03045* expression groups. **D)** Volcano plot showing upregulated and downregulated genes in high vs. low *LINC03045* expression groups. **E)** KEGG pathway analysis. **F)** Gene ontology analysis. **G)** Venn diagram illustrating the intersection of differentially expressed genes in our *in vitro LINC03045* KD and *in vivo LINC03045* TCGA gene expression profiles.

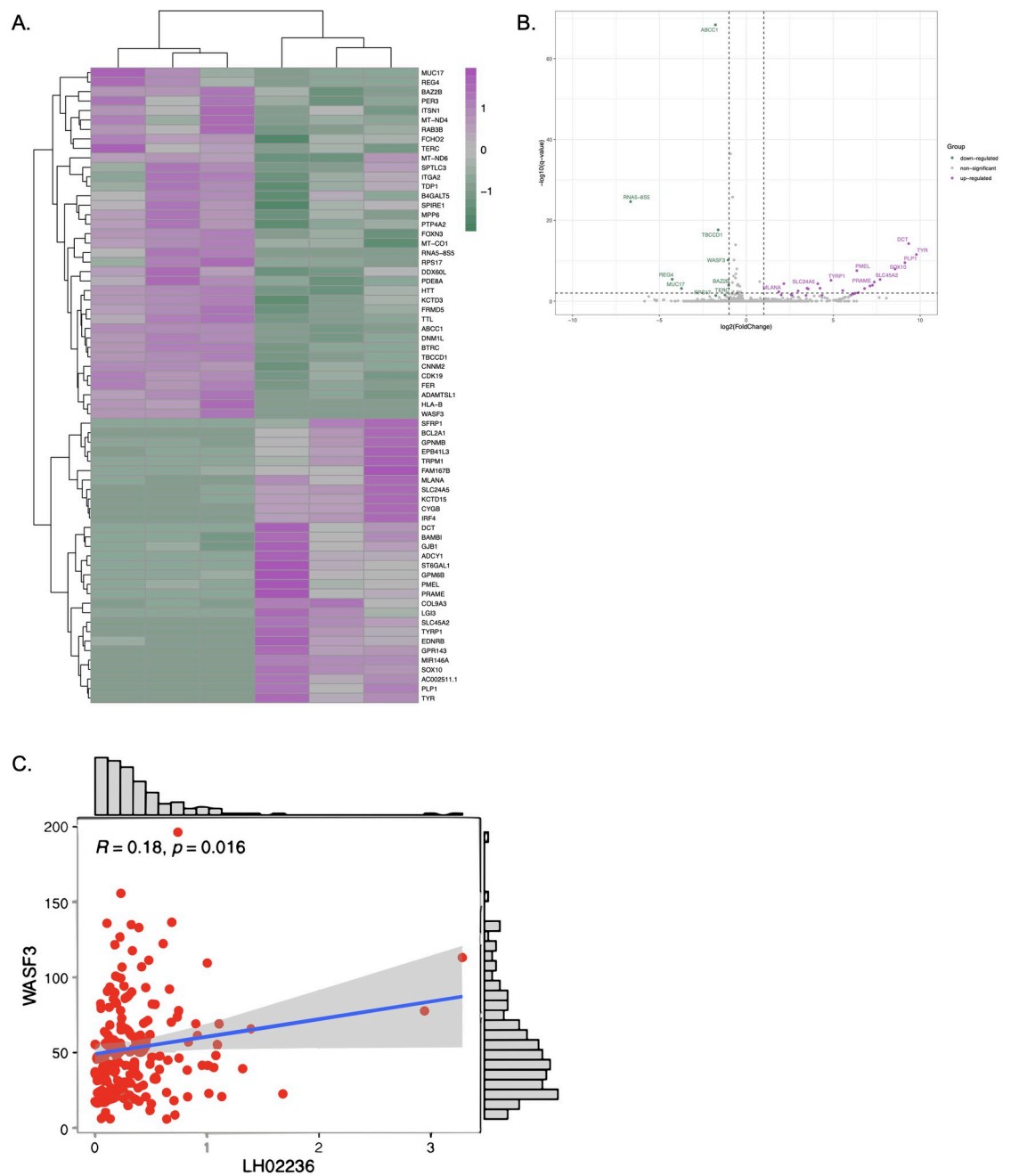

**Fig 6. RNAseq analysis of *LINC03045* KD cells. A)** Heat map showing differential expression of genes with *LINC03045* KD (right) compared to control (left). **B)** Volcano plot showing differential gene expression with *LINC03045* KD. **C)** Analysis of *WASF3* correlation with *LINC03045*.

derived cell line, USC02. Upon ASO-mediated *LINC03045* KD, we once again noted a transcriptional decrease in *LINC03045* (82.3% decrease, p = 0.001) and *WASF3* (68.8% decrease, p = 0.02) (Fig 8A). Concurrent ASO-mediated *LINC03045* KD and *WASF3* overexpression demonstrated a transcriptional decrease in *LINC03045* (51.9% decrease, p = 0.001) as well as a dramatic transcriptional increase of *WASF3* (14,816.4% increase, p = 0.008) (Fig 8A). Assessment of cell invasion via Matrigel assays reproducibly demonstrated the statistically significant

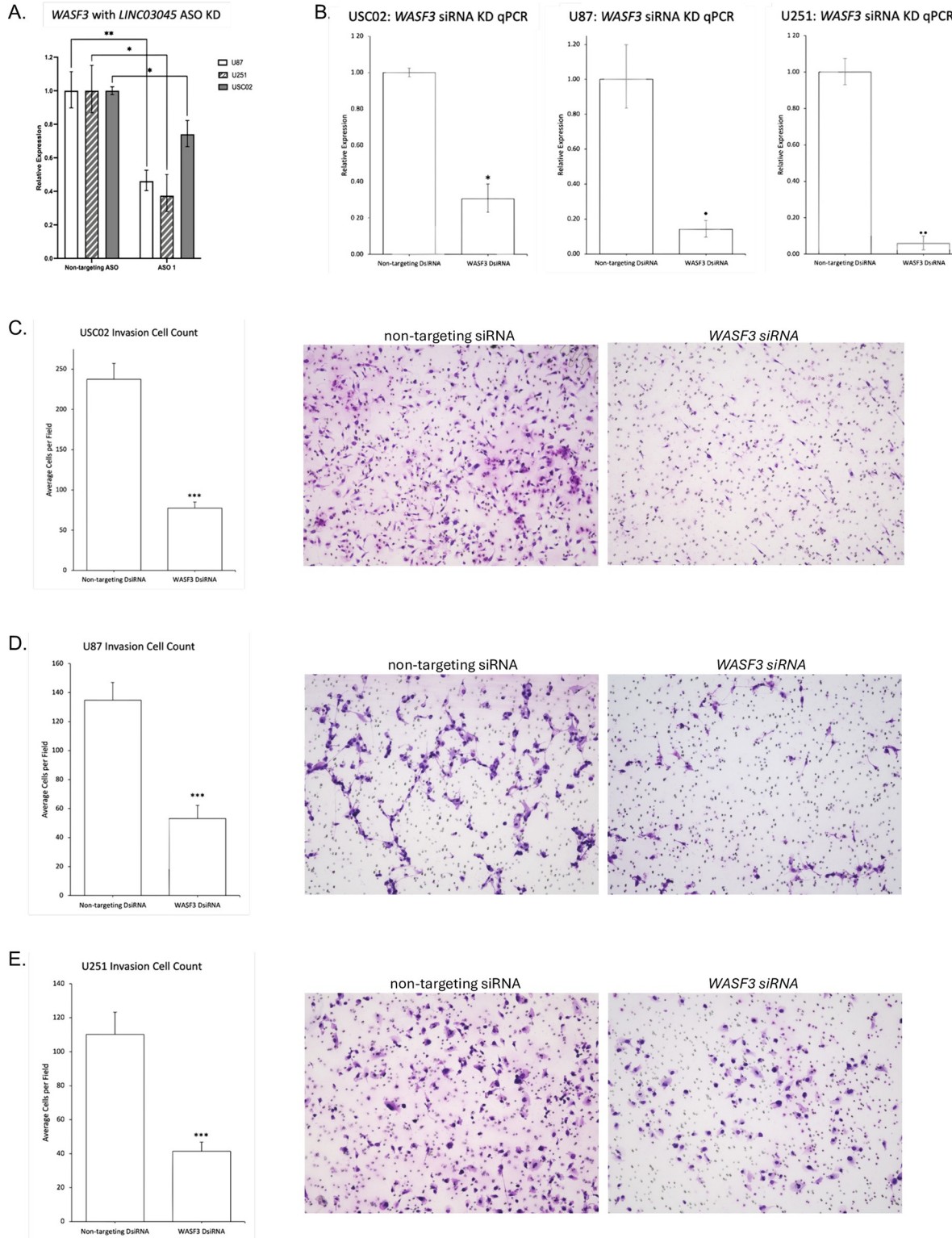

**Fig 7. WASF3 affects invasion. A)** qPCR of *WASF3* after antisense oligonucleotide (ASO) KD of *LINC03045* in USC02, U87, and U251 cells. All expression levels are normalized to the housekeeping gene *RPLP0*. KD cells were compared to the non-targeting ASO control. Data are expressed as mean ± SD for replicates. *p ≤ 0.05, **p ≤ 0.01, ***p ≤ 0.001 **B)** qPCR of dicer-substrate short interfering RNA (siRNA) KD of *WASF3* in USC02, U87, and U251 cells. All expression levels are normalized to the housekeeping gene *RPLP0*. KD cells were compared to the non-targeting siRNA control. Data are expressed as mean ± SD for replicates. *p ≤ 0.05, **p ≤ 0.01, ***p ≤ 0.001 **C–E)** Representative

images of invasion assay through a Matrigel-coated Boyden chamber after KD of *WASF3* in USC02 (C), U87 (D), or U251 (E) cells by siRNA. The bar graph represents the number of invaded cells per field counted, with KD cells compared to the non-targeting siRNA. Data are expressed as mean ± SEM of three independent experiments with 9 fields imaged per experiment. *p ≤ 0.05, **p ≤ 0.01, ***p ≤ 0.001.

decrease in invasion relative to control for ASO-mediated *LINC03045* KD only (54.8% decrease, p = 0.002), and when "rescuing" *LINC03045* KD tumor cells with concurrent *WASF3* overexpression, cells demonstrated a statistically significant increase in invasive capacity compared to *LINC03045* KD only cells (234.5% increase, p = 0.002) (Fig 8B and 8C). However, when compared to *WASF3* overexpression only, the rescue cells showed no statistically significant difference in invasion (30.8% decrease, p = 0.08) (Fig 8B and 8C).

## Discussion

To efficiently identify novel lncRNAs in GBM that affect the invasive ability of malignant cells and determine possible therapeutic targets, we developed a CRISPRi-based invasion screen. Here we report the first large-scale CRISPRi screen on lncRNAs implicated in glioblastoma invasion. Our results identified *LINC03045* as significantly associated with *in vitro* glioma cell invasion across 3 tumor lines. Multiple methods of lncRNA knockdown further validated the functional role of *LINC03045* via CRISPRi-based and ASO-based knockdown in all three lines. Patient gene expression analysis of TCGA, and GTEx raw transcriptional data revealed that *LINC03045* expression was associated with tumor grade and patient survival, as well as widespread changes in global gene expression data, including oxidative phosphorylation and RNA processing. Finally, gene expression analysis following *LINC03045* KD revealed that *WASF3*, a

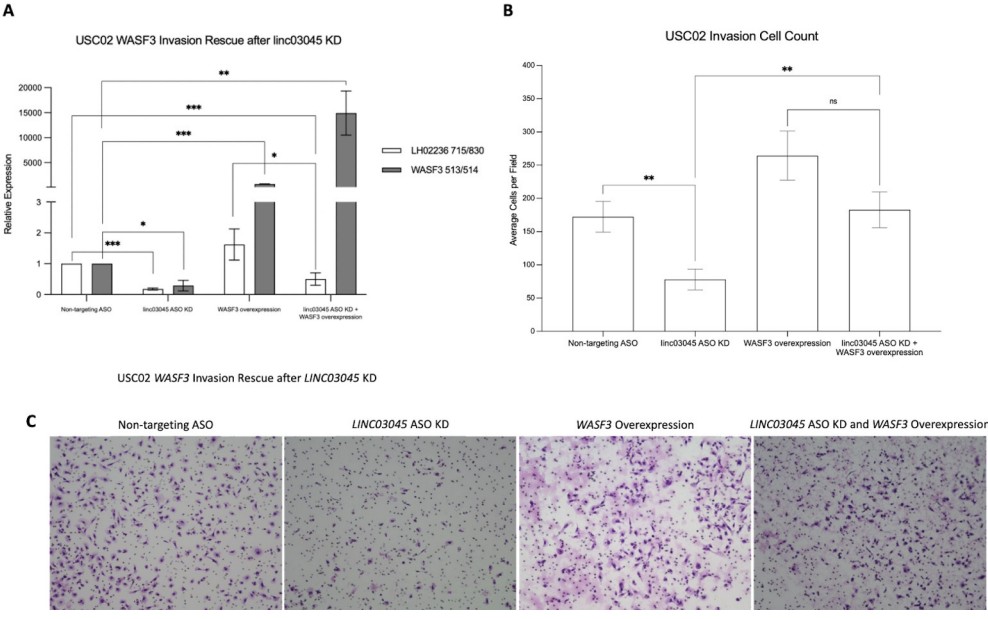

**Fig 8. *WASF3* is a downstream target of *LINC03045*. A)** qPCR of *LINC03045* and *WASF3* after antisense oligonucleotide (ASO) KD of *LINC03045* and *WASF3* overexpression in USC02 cells. All expression levels are normalized to the housekeeping gene *RPLP0*. KD cells were compared to the non-targeting ASO control. *p ≤ 0.05, **p ≤ 0.01, ***p ≤ 0.001 **B)** Bar graph represents the number of invaded cells per field, with KD cells compared to the non-targeting ASO and concurrent KD and *WASF3* overexpression cells compared to *WASF3* overexpression only. Data are expressed as mean ± SEM of three independent experiments with 9 fields imaged per experiment. **p ≤ 0.01 **C)** Representative images of invasion assay through a Matrigel-coated Boyden chamber after ASO KD of *LINC03045* and *WASF3* overexpression in USC02 cells.

factor evaluated for tumor cell invasion in multiple studies, could be a possible mediator of *LINC03045* function [30,31]. Mechanistic studies supported the role of *WASF3* as a lncRNA mediator, with *WASF3* KD paralleling invasion loss seen with *LINC03045* KD in both 2D Matrigel and 3-dimensional invasion modles. Finally, *WASF3* rescue experiments confirmed that *WASF3* overexpression restores loss-of-invasion from lncRNA KD.

The CRISPRi invasion screen of 2,307 loci identified 45 hits that affect GBM invasion. Upon evaluation of these candidate hits, *LINC03045*, a previously unannotated and poorly studied lncRNA, was noted to be significantly associated with invasion. Functional genomic methods such as CRISPR (and CRISPRi) screens have gained significant popularity in recent years due to the ability demonstrated here to quickly and efficiently identify potential therapeutic targets [14,19]. While prior manuscripts have notably focused on the role of coding genes affecting glioma invasion [32], the body of literature characterizing the role of lncRNAs in GBM pathogenesis has only just begun to grow in recent years and have primarily focused on the role of lncRNAs in GBM proliferation and progression [33–37]. Prior studies on lncRNAs in GBM invasion, however, remain scarce, and our study uniquely addresses the role of lncRNAs specifically in GBM invasion, suggesting that noncoding RNAs play a significant functional role in this phenotype. Knockdown of our candidate lncRNA via multiple methods, including via both CRISPRi and ASO, revealed similar decreases in tumor invasion while not affecting tumor proliferation. While our knockdown models demonstrated 40–70% decrease in lncRNA expression, we did notice relatively dramatic phenotypic changes in invasion of over 80% in the Matrigel invasion after a knockdown of less than 40%. Part of the observed effect is likely variability. However, essential genes in certain processes have been previously noted to show exaggerated phenotypes with small changes in gene expression, which could play a role in the observed changes in our results [38]. In addition, because lncRNAs have been commonly characterized as extremely abundant and specific to a small subset of cells [39], another possible explanation is that gene expression was particularly abundant in an aggressive, invading, lncRNA-rich subpopulation is capable of demonstrating invasion at 24 hours during the assay. Under this assumption, it is possible a more modest 40% knockdown of the highest expressing cells can more significantly affect the 24 hour invasion phenotype.

LncRNAs represent challenging experimental and therapeutic targets. CRISPR/Cas9 or CRISPRi-based drugs carry inherent clinical challenges of non cell-specific effects for treating diseases. ASOs are single-stranded nucleotide strands that are taken up freely by cells to form heteroduplexes with the complementary target RNA, designating them for degradation, and can offer a potential treatment option. since they are single-stranded nucleotide strands that are taken up freely by cells to form heteroduplexes with the complementary RNA. Clinically, ASO-mediated therapies are currently approved and being utilized for treating diseases like spinal muscular atrophy [40]. Thus, we also successfully utilized ASOs as a confirmatory second knockdown method and therapeutic medium to knock down *LINC03045* and evaluate subsequent effects.

Notably, our experimental data was also confirmed to be clinically relevant, as TCGA and GTEx analysis showed that *LINC03045* expression is higher in GBM (grade 4) than in LGG (grades 2–3), and higher in LGG when compared to normal cortical tissue. In addition, Kaplan-Meier survival curves for all gliomas (LGG and GBM) showed patients with higher *LINC03045* expression were associated with decreased survival. After adjusting for grade and IDH-status, higher *LINC03045* expression was significantly associated with lower survival only in the LGG cohort that included grades 2 and 3. The survival analysis for this LGG cohort included both grades 2 and 3 because the TCGA defines LGG as these two grades, and a portion of the LGG dataset did not further distinguish grade between them [41]. Notably, grade 2 and 3 gliomas within this cohort did not significantly differ in linc03045 expression.

However, we also conducted analysis of Grade 2 gliomas IDH-mut tumors alone when excluding the 20 patients with uncertain grade 2/3 pathology. Notably, in this cohort, the sample size was significantly reduced (possibly underpowered), though the trends in the survival curves remained similar between Grade 2 IDH-mut curves and Grade 2/3 curves. However, *LINC03045* expression was not associated with survival in this cohort (p = 0.18, Fig 2).[42]. Further, *LINC03045* also did not appear to be significantly associated with survival when strictly in the GBM cohort (stage 4) with adjustment for IDH-status. Notably, it appears that in grade 4 gliomas, the vast majority of samples express linc03045 >1.46, which is greater than the 75th percentile of linc03045 expression in all other tumor grades. It may be that above a significant threshold of gene effect, or with a the more limited survival of grade 4 tumors, the gene expression in Grade 4 tumors is less significant to survival. Nonetheless, the enhanced expression of *LINC03045* in GBM in comparison to normal brain cortex and LGG suggests that it has an association with GBM aggressiveness and higher grade. Prior literature has shown that lncRNA expression has been associated with GBM outcome in multiple studies and has been explored in other lncRNAs as a biomarker for prognosis [11,43]. Thus, further study of these findings may reveal the role of *LINC03045* expression in resected patient tumors as a prognostic biomarker.

While lncRNA-based prognostic and diagnostic markers and lncRNA-based therapeutics described above are still in their experimental stages, the study of lncRNAs often offers insight into mechanisms regulating tumor phenotypes such as, in our study, invasion. When evaluating the intersection between differential gene expression in *LINC03045* KD *in vitro* and *LINC03045* expression changes *in vivo* (in the TCGA), we noted an overlap of 16 out of the 36 differentially expressed genes *in* vitro, with significant alterations in GO pathways in cell shape and neural crest cell migration (S1 Table). In prior literature, implicated overlapping genes, including *WASF3*, *EDNRB*, *SOX10*, *BAMBI* have been implicated in processes including tumor progression, migration, and invasion [44,45]. Of particular interest in our study was *WASF3*. Here, our RNA-seq analysis following *LINC03045* KD revealed that *WASF3* is a gene that was reproducibly downregulated with *LINC03045* knockdown, while a correlation analysis of patient TCGA/GTEx sequencing data revealed a significant positive correlation between *LINC03045* and *WASF3*. Additionally, the GO term for cytoskeleton organization (GO: 0007010) process was enriched for *WASF3*, which is consistent with prior literature that describes cytoskeletal reorganization as commonly associated with tumor invasion [46]. *WASF3*, a Wiskott-Aldrich syndrome protein family member, has been implicated in multiple studies as a regulator of the actin cytoskeleton and in cancer invasion, which is again consistent with our results [47]. Studies have shown that it is a scaffolding protein that is essential in coordinating metastatic signaling complexes, promoting invasion and metastasis through activation of leading edge membrane structures. *WASF3* has been shown to be associated with invasion in several cancers, including breast and ovarian cancer, and is inversely correlated with overall progression-free survival [48]. Notably, *WASF3* is a critical component of the JAK-STAT3 pathway. STAT3 promotes *WASF3* expression while JAK2 independently activates it, and this process is necessary for invasion [30,31]. *WASF3* overexpression also activates *NFkB* and *ZEB1* and also promotes invasion by regulating genes involved in metastasis. While we have noted that *WASF3* could be a mediator of *LINC03045* function, lncRNAs typically regulate transcriptional activity of target genes by acting as scaffold or cofactor for transcription factors. Future studies are needed to characterize the mechanistic role by which *LINC03045* regulates *WASF3* gene expression in the context of its known role in the JAK--STAT3 pathway.

*LINC03045* knockdown was futher associated with widespread gene expression changes potentially affecting tumor invasion, as noted in KEGG pathway analysis. Pathway analysis

found that genes involved in oxidative phosphorylation pathways were most strongly associated with *LINC03045*. Prior studies have found that aberrations in glycolysis and oxidative phosphorylation could contribute to GBM aggressiveness and invasion and that an acidic and hypoxic tumor microenvironment can promote tumor invasion by changing glycolytic flux, thus promoting local protumorigenic inflammatory responses [49].

Other notable processes identified to be associated with *LINC03045* from KEGG analysis were ribosomal processes and spliceosomal processes. A prior study found that ribosomal proteins are associated with enzymes that promote proliferation and invasion of GBM cells as well as apoptotic processes [50,51]. Another previous study found that spliceosome assembly and alternative splicing could contribute to GBM invasion and the GBM mesenchymal phenotype [52].

Our study does carry limitations. Although CRISPR/Cas9 is a robust tool to target lncRNAs, major limitations include risk of affecting adjacent genes, including those that overlap with the lncRNA gene or elements in the lncRNA locus that may regulate other genes [33]. Notably, our gene expression analysis following *LINC03045* KD in patient cell lines revealed that no coding genes, with the closest one residing over 1kB from *LINC03045*, were significantly changed following lncRNA KD. Another limitation lies in the challenge of screening a complex phenotype such as invasion. We attempted to circumvent this limitation by performing screens over the course of 24 hours, a time period during which pilot studies (and prior studies [14,19,32]) showed adequate invasion. Furthermore, our MTT assays suggested no significant alterations in tumor proliferation during this screening time period. In addition, our study initially screened CRISPRi lncRNA knockdowns within a commercial tumor cell line, U87. The 24-hour screen of a patient-derived cell line proved challenging due to difficulties achieving stable transfection of both the dCas9 and sgRNA plasmids. While we demonstrated consistent phenotype and downstream changes in multiple cell lines, including a recently derived patient tumor line, and included a 3D invasion assay as well, further comprehensive screens may include screening of more patient lines. Additionally, given the technical challenges of conducting long-term lentiviral knockdown of lncRNAs *in vivo*, including target specificity and the instability of lentiviral transduction of the *LINC03045* knockdown, we have chosen to utilize 3D invasion assays as an approximation of *in vivo* behavior, which allowed for stable, continuous knockdown of *LINC03045* via ASO [53–55]. However, a true *in vivo* knockdown study of *LINC03045* is needed in further studies to corroborate our results, since *in vivo* models are still the most accurate method for studying tumor behavior and environmental interactions [56]. Furthermore, though *LINC03045* appears to modulate *WASF3* to impact GBM invasion, we did not perform formal mechanistic studies involving co-localization or RNA pulldown. Potential hypotheses for a possible mechanism include functioning as a microRNA sponge, lncRNA interaction with transcriptional regulators, or functioning as a guide or scaffold for transcription factors of either *WASF3* or its upstream regulators. However, the transcriptional regulation of WASF3 still remains relatively understudied, and elucidating the mechanism behind these relationships would require extensive further research.

After utilizing a large-scale functional CRISPRi invasion screen, *LINC03045* was identified to regulate *in vitro* invasion of multiple GBM cell lines. Subsequent evaluation of patient clinical data reveals significant associations between *LINC03045* and tumor grade, as well as patient survival. Analysis of patient data, tumor cell lncRNA knockdown, and *WASF3* knockdown reveal that *WASF3* mediates the *LINC03045* tumor effect. While future studies must characterize the specific mechanism by which *LINC03045* modulates *WASF3* expression, as well as the role of *LINC03045* using *in vivo* models, our results show that *LINC03045* has an important role in GBM invasion.

## Methods

### Cell culture

U87 cells (ATCC, HTB-14; RRID:CVCL_0022), HEK293T cells (ATCC, CRL-3216; RRID: CVCL_0063), and U251 cells (Sigma-Aldrich, 09063001-1VL; RRID:CVCL_0021) were maintained in D10 medium, which contained high glucose Dulbecco's modified Eagle's medium (DMEM) (Gibco, 11995–081) supplemented with 10% fetal bovine serum (FBS) (Seradigm, 97068–085) and 1X antibiotic-antimycotic (Gibco, 15240–062). USC02 cells, a primary glioma stem cell (GSC) line, were a gift from Thomas C. Chen [29] and maintained in cancer stem cell (CSC) medium containing DMEM/F-12 (Gibco, 11320–033) supplemented with 1% B-27 (Life Technologies, 12587–010), 1X antibiotic-antimycotic, and 20 ng/mL epidermal growth factor (EGF) (Peprotech, AF-100-15) and basic fibroblast growth factor (bFGF) (Peprotech, 100-18B). All cells were kept in a humidified incubator at 37˚C with 5% $CO_2$ and confirmed to be mycoplasma free with the LookOut Mycoplasma PCR Detection Kit (Sigma-Aldrich, MP0035) immediately prior to a large-scale screen and at least once every two months.

### CRISPRi screens

U87 cells expressing dCas9-KRAB-BFP were generated by lentiviral infection. pHR-SFFV-dCas9-BFP-KRAB was a gift from Stanley Qi & Jonathan Weissman (Addgene plasmid #46911; http://n2t.net/addgene:46911; RRID:Addgene_46911) [57]. Fluorescence associated cell sorting (FACS) was used 48 hours post-transduction to isolate cells stably expressing BFP (U87+dCas9-KRAB).

The U87-specific CRISPR inhibition (CRISPRi) library, derived from the human CRISPRi non-coding library (CRinCL) (Addgene #86542, a gift from Jonathan Weissman) by selecting sub-libraries that targeted all expressed lncRNAs in U87 [19], was used to generate the sgRNA pool used for screening. TransIT-Lenti (Mirus, 6600) was used to package the plasmids for transfection in HEK293Ts and viral production was allowed to occur for 48 hours before harvesting. Filtered viral supernatant was then added to U87+dCas9-KRAB cells with polybrene (Sigma-Aldrich, TR-1003-G) at a final concentration of 4 μg/mL at an MOI < 0.3. Two days after infection, cells were treated with 0.75 μg/mL puromycin (Gibco, A1113803) for 3 days, allowed to recover for one day without puromycin, and used for assays the next day ($T_0$), with cells being maintained at a coverage of 1000X. Two independent screen replicates were conducted, with cells collected at $T_0$ and at the endpoint, and processed for sequencing on Illumina HiSeq 4000 and analyzed as previously described [19]. A scatterplot of invasion phenotype of sgRNA of two independent replicates was generated, with the x and y-axes representing different screen replicates (S2 Fig). Analysis output denoted p-value and effect sizes for lncRNAs targeted in the screen. A specific screen score, incorporating these values, was calculated via x -log10 p value relative to negative internal control sgRNA targets as previously described [19,22]. These were plotted via volcano plot for selection of highest screen scores.

### Cell invasion assay

Boyden chambers comprised of polyethylene terephthalate with 8 μm pore size and coated in a layer of Matrigel (Corning, 354481) were used to test invasion of cells. U87 screen cells, prepared as described above, were seeded at a density of 5 x $10^5$ cells. For individual knockdown validation, U87 cells (5 x $10^5$), U251 cells (1.25 x $10^5$), or USC02 cells (2 x $10^5$) were seeded in the top chamber with serum-free medium while the lower chamber contained D10 medium as the chemoattractant. Chambers were then incubated for 24 hours in a humidified incubator at

37°C with 5% $CO_2$. For deep sequencing following screen, cells were collected from the top chamber by trypsinization.

To count cells that had invaded when evaluating individual knockdown in validation studies, a cotton swab was used to remove cells remaining in the upper chamber and cells on the underside were fixed and stained with Hemacolor Stain Set (Sigma-Aldrich, 65044). Photos were taken on a Keyence, BZ-X800, with 9 fields per chamber, and cells counted with Image*J* Software. Experiments were performed in triplicates and independent t-test used to evaluate statistical significance.

## 3D spheroid invasion assay

Cultrex spheroid invasion assay (3500-096-K) was used to model 3D invasion of glioma-spheres into surrounding invasion matrix. Control (untransfected) U87) cells, and U87 cells transfected with *LINC03045* ASO or WASF3 siRNA were plated in 50ul of 1X spheroid-forming matrix diluted in serum-free DMEM/F12 medium in a 96-well round-bottom ultra-low attachment plate on ice. Plated cells were immediately centrifuged at 200g for 3 min at room temperature to concentrate the cells in the center of the welland enhance gliomasphere formation. After incubation at 37°C for 72h, the spheroid-containing plate was equilibriated on ice for 15 mins, followed by addition of 50ul 1X invasion matrix per well, and the plate was centrifuged at 300g for 5min at 4°C. After incubation for 1 hour at 37°C, 100ul of chemoattractant medium (10%FBS final concentration) was added to each well, and cells were allowed to invade into the surrounding matrix. Serum-mediated invasion was monitored over 48 hours, and wells were imaged at 2x magnification on the Keyence, BZ-X800. Invasion area was manually outlined, and measured using the Image*J* Software. Invasion from ASO and siRNA-transfected spheroids was compared to invasion from untransfected control U87 spheroids. Experiments were performed in triplicates, and statistical significance between groups was evaluated using independent t-tests.

## Validation of screen candidates via knockdown

To clone the sgRNAs for individual validation, oligo pairs containing the sgRNA protospacer and adjoining BstXI and BlpI cloning sites were annealed and ligated into the sgRNA plasmid backbone pU6-sgRNA EF1Alpha-puro-T2A-BFP (Addgene plasmid #60955; http://n2t.net/addgene:60955; RRID:Addgene_60955, a gift from Jonathan Weissman) [22]. A non-targeting control (*GAL4*) was also subcloned into the aforementioned backbone. The final sgRNA sequences used were: *GAL4* 5'-GAACGACTAGTTAGGCGTGTA-3'; *LINC03045* sgRNA 1 5'-GTTTAACCAACGGGTTATTT-3'; *LINC03045* sgRNA 2 5'-GTTTGTTTTATTAGTAAAAG-3'.

## RT-qPCR

TRIzol Reagent (Invitrogen, 15596–018) or the RNeasy Plus Mini Kit (Qiagen, 74134) was used to extract total RNA 24 or 48 hours following ASO or *WASF3* overexpression plasmid (*WASF3* NM_006646) Human Tagged ORF Clone (Origene, RG216156) transfection, or at $T_0$ for CRISPRi. Genomic DNA removal and cDNA synthesis were completed in the same tube with the iScript gDNA Clear cDNA Synthesis Kit (Bio-Rad, 1725034). qPCR was performed with the iTaq Universal SYBR Green Supermix (Bio-Rad, 1725121) on a Roche LightCycler 96 instrument. Experiments were performed in triplicates and independent t-test used to evaluate statistical significance. Primers used for qPCR were: *RPLP0* forward 5'- CATATCCGGGG-GAATGTGGG-3'; *RPLP0* reverse 5'- AGCAGCTGGCACCTTATTGG-3'; *LINC03045* forward 5'- CCCATCCAAGAAGTTTGCAG-3'; *LINC03045* reverse 5'-

GCCACTGTCTTTCTGCTTCC-3'; *WASF3* forward 5'-TCAACAGTGGAAGAGGTCTCA C-3'; and *WASF3* reverse 5'- TTCAGAGGCGGTGGCTTATC-3'.

## Cell proliferation assay

3-(4,5-dimethylthiazol-2-yl)-2,5-diphenyltetrazolium bromide (MTT) assays were used to measure cell proliferation during the time points cells would be undergoing invasion assays. Cells (USC02, U87, and U251) were seeded in 96-well plates (5 x $10^3$ cells/well) in 100 μL of complete media and the next day incubated with 10 μL of 5 mg/mL tetrazolium dye (Invitrogen, M6494) at 37°C with 5% $CO_2$ for 2 hours. The solution was then removed, and the cells incubated for 10 minutes at room temperature with 100 μL of dimethyl sulfoxide (Sigma-Aldrich, D8418). The optical density values at 570 nm and 650 nm were detected with a Thermo Scientific Varioskan LUX microplate reader. Experiments were performed in triplicates and independent t-test used to evaluate statistical significance.

## Antisense oligonucleotides

2'-O-methoxyethyl (2'-MOE) modified antisense oligonucleotide (ASO) gapmers were designed against linc03045 by Integrated DNA Technologies. ASOs were transfected at a final concentration of 25 nm with *Trans*IT-siQUEST Transfection Reagent (Mirus, MIR 2110) or the *Trans*IT-X2 Dynamic Delivery System (Mirus, MIR 6003) following the manufacturer's protocol. U87 and U251 cells were transfected with a either a non-targeting ASO or a linc03045 ASO for 24 hours while USC02 cells were transfected for 48 hours. ASO target sequences were: non-targeting 5'-/52MOErG/*/i2MOErC/*/i2MOErG/*/i2MOErA/ */i2MOErC/*T*A*T*A*C*G*C*G*C*A*/i2MOErA/*/i2MOErT/*/i2MOErA/*/i2MOErT/ */32MOErG/-3'; *LINC03045* ASO 1 5'-/52MOErA/*/i2MOErT/*/i2MOErC/*/i2MOErG/ */i2MOErT/*T*T*C*T*C*G*G*C*C*T*/i2MOErT/*/i2MOErT/*/i2MOErT/*/i2MOErG/ */32MOErG/-3'; and *LINC03045* ASO 2 5'-/52MOErT/*/i2MOErA/*/i2MOErG/*/i2MOErG/ */i2MOErC/*A*C*A*A*G*C*C*A*C*C*/i2MOErG/*/i2MOErT/*/i2MOErT/*/i2MOErT/ */32MOErA/-3'.

## RNA-seq sample preparation and data analysis

U87 cells were transfected with either a non-targeting or *LINC03045* ASO for 24 hours as described above, spun down into a pellet, and flash frozen. Cell pellets underwent RNA extraction, library generation by Poly(A) selection, and sequencing on an Illumina HiSeq with the 2x150 bp configuration (GENEWIZ, Azenta Life Sciences).

Pair-end raw sequencing reads were processed as follows: Firstly, we adapted Trim Galore [58] to remove adaptors and low-quality reads. Cleaned reads with length greater than 20 were mapped to the human hg19 reference genome by using tophat [59] with default parameters. To quantify the expression of *LINC03045*, GENCODE [60] release 19 was downloaded from the ensemble database with subtle modifications. We added the coordinates of *LINC03045* to the GENCODE annotation database. Then, featureCount [61] was employed to count reads in individual genes. A matrix containing the read counts was subjected to the DEseq2 [62] package for differential expression genes analysis. In addition, Reads Per Kilobase Million (RPKM) was used to quantify the expression level of each genes. Genes with an absolute value of log2 fold change greater than 1 and a Benjanmini-Hochberge adjusted p-value less than 0.05 were considered differentially expressed for further validation.

### Dicer-substrate short interfering RNA

Dicer-substrate short interfering RNAs (siRNAs) were designed against *WASF3* by Integrated DNA Technologies. siRNAs were transfected at a final concentration of 10 nm with *Trans*IT-siQUEST Transfection Reagent following the manufacturer's protocol. U87, U251, and USC02 cells were transfected with either a non-targeting ssiRNA or a *WASF3*-targeting siRNA.

### TCGA and GTEx analysis

RNA-seq alignment data (BAM files aligned to GRCh38) of patient samples and normal brain cortical tissue were acquired through the Database of Genotypes and Phenotypes (dbGaP), specifically Genotype-Tissue Expression (GTEx) for normal tissue (n = 104) and The Cancer Genome Atlas (TCGA) for GBM (n = 169) and lower grade glioma (LGG) (n = 169). The Samtools package was used to filter the BAM files by flag status to retain read alignments that mapped in proper pairs, after which expression read counts were tabulated for the positional coordinates of *LINC03045* (chr10:92658250–92661763). The expression counts were normalized to total read alignments on chromosome 10 to obtain counts per million (CPM) values. Gene expression distribution shifts amongst the three cohorts were observed using histograms. DESeq2 was used to calculate to calculate the significance of *LINC03045* differential expression between the glioma datasets and control, against a backdrop distribution of all other chromosome 10 genes. Information on WHO grade and molecular subtype (e.g. IDH status) of the GBM and LGG cohorts from TCGA were obtained from a prior study[63].

Kaplan-Meier (KM) survival curve and log-rank test were performed for all gliomas (LGG and GBM), LGG and GBM. Cox regression was conducted using expression levels as a continuous variable. Variables of IDH status, MGMT methylation, and 1p/19q codeletion, when available, were utilized in multivariate Cox regression analysis. None of the variables were acting as confounders for the gene expression and survival. To avoid overadjustment we present results using gene expression alone. Linearity assumption and proportional hazard assumption was evaluated using Martingale residuals. For *LINC03045*, linearity was violated, and the model was built using two linear splines with the knot at 2.5 CPM. The knot position was tested by modeling 2.0 to 3.0 by 0.1 increment. 2.5 CPM was chosen as the lowest Akaike information criterion (AIC) value. Significance level was 0.05, 2-sided. All analyses were performed using SAS 9.4 (SAS Institute Inc., Cary, NC, USA.).

For GBM patiens RNA-seq data from TCGA, raw reads were processed as previously described in RNA-seq analysis section, with modifications. Briefly, reads were trimmed using Trim Gare [58] and then mapped to the hg19 reference genome with STAR [64]. The gene expression was quantified in TPM at gene level using GENCODE [60] version 19 gene annotation, including the postion of *LINC03045*. To obtain the genes that are highly correlated with *LINC03045*, the Pearson correlation coefficients were calculated between the target lncRNA and any other genes. A threshold of p-value less than 0.05 was reported as remarked genes. Significant positively correlated genes (correlation coefficient >0) and negatively correlated genes (correlation coefficient <0) were submitted to the online DAVID [65] database for Gene Ontology (GO) enrichment analysis, repectively. In addition, patients were divided into high and low groups, according to the *LINC03045* expression value at the upper 25 percent and lower 25 percent. These two groups were loaded into the Limma [66] package for differential expressed genes detection. Differential expression analyses (DEGs) were also carried out for enriched GO analysis. Gene Set Enrichment Analysis (GSEA) was performed to enrich pathways using the clusterProfiler [67] package for statistical analysis and visualization.

## Supporting information

**S1 Fig. Annotated volcano plot of invasion phenotype and negative $\log_{10}$(Mann-Whitney p-value).** Screen replicates were averaged and the top 3 sgRNAs for each lncRNA compared to non-targeting controls were used to determine screen hits. The dashed lines represent thresholds to determine screen hits.
(TIF)

**S2 Fig. Scatterplot of invasion phenotype of sgRNA of two independent replicates.** A positive phenotype indicates knockdown of a lncRNA caused a decrease in invasion, whil a negative phenotype indicates knockdown of a lncRNA caused an increase in invasion. The X and Y-axes represent different screen replicates.
(TIFF)

**S3 Fig. Preliminary results of the ZNF215-AS1 candidate. A)** qPCR of CRISPRi KD of the top two sgRNAs for ZNF215-AS1 in U87 cells. All expression levels normalized to the housekeeping gene RPLP0. KD cells compared to the non-targeting sgRNA. Data expressed as mean ± SD for replicates. **B)** Representative images of invasion assay through a Matrigel-coated Boyden chamber after KD of ZNF215-AS1 by CRISPRi KD in U87 cells. The bar graph represents the number of invaded cells per field counted, with KD cells compared to the non-targeting sgRNA. Data are expressed as mean ± SEM of three independent experiments with 9 fields imaged per experiment. **C)** An MTT assay showing no change in proliferation after KD of ZNF215-AS1 or LH02236 by CRISPRi in U87 cells. KD cells compared to the non-targeting sgRNA. Data are expressed as mean ± SD. $^*p \leq 0.05$, $^{**}p \leq 0.01$, $^{***}p \leq 0.001$ **D)** Boxplots of pairwise comparisons of ZNF215-AS1 expression in normal human brain cortex to low grade glioma (LGG) and glioblastoma (GBM) patient samples. **E)** Kaplan-Meier survival curves of ZNF215-AS1 for gliomas (stages 1–4) with expression quantified by normalized counts per million (CPM). Position coordinates were used to identify both genes in RNA-seq data. Expression was normalized to standardized genes on the same chromosome. Censored data indicates patients lost to follow-up. $^*p \leq 0.05$, $^{**}p \leq 0.01$, $^{***}p \leq 0.001$
(TIFF)

**S4 Fig.** qPCR confirmation of *LINC03045* KD (A) and *WASF3* KD (B) in U87 cells relative to control untransfected U87 cells, after transfection with *LINC03045* ASO or *WASF3* siRNA respectively. All data were normalized to *RPLPO* housekeeping gene. Data are expressed as mean ± SEM for replicates. $^*p \leq 0.05$, $^{**}p \leq 0.01$, $^{***}p \leq 0.001$. (C) Quantification and representative images of U87 spheroid invasion into surrounding invasion matrix after ASO-mediated knockdown of *LINC03045* compared to control untransfected U87 spheroids. Data are expressed as mean ± SEM for replicates. $^*p \leq 0.05$, $^{**}p \leq 0.01$, $^{***}p \leq 0.001$. (D) Quantification and representative images of U87 spheroid invasion into surrounding invasion matrix after siRNA-mediated knockdown of *WASF3* compared to control untransfected U87 spheroids. Data are expressed as mean ± SEM for replicates. $^*p \leq 0.05$, $^{**}p \leq 0.01$, $^{***}p \leq 0.001$. Spheroid images taken at 2x magnification.
(TIF)

**S1 Table. Gene Ontology analysis of the relationship between WASF3 and enriched pathways in *in vivo* gene expression.**
(TIFF)

## Author Contributions

**Conceptualization:** Josh Neman, Yali Dou, Frank J. Attenello.

**Data curation:** Kathleen Tsung, Kristie Q. Liu, Krutika Deshpande, Yong-Hwee Eddie Loh, Li Ding, Wentao Yang.

**Formal analysis:** Kathleen Tsung, Kristie Q. Liu, Jane S. Han, Krutika Deshpande, Tammy Doan, Yong-Hwee Eddie Loh, Li Ding, Wentao Yang, Frank J. Attenello.

**Funding acquisition:** Li Ding, Frank J. Attenello.

**Investigation:** Kathleen Tsung, Kristie Q. Liu, Jane S. Han, Krutika Deshpande, Tammy Doan, Yong-Hwee Eddie Loh, Li Ding, Wentao Yang.

**Methodology:** Kathleen Tsung, Kristie Q. Liu, Jane S. Han, Krutika Deshpande, Tammy Doan, Yong-Hwee Eddie Loh, Li Ding, Wentao Yang, Josh Neman, Yali Dou, Frank J. Attenello.

**Project administration:** Kathleen Tsung, Kristie Q. Liu, Krutika Deshpande, Josh Neman, Frank J. Attenello.

**Resources:** Krutika Deshpande, Yali Dou, Frank J. Attenello.

**Software:** Yong-Hwee Eddie Loh, Li Ding, Wentao Yang.

**Supervision:** Josh Neman, Yali Dou, Frank J. Attenello.

**Validation:** Kathleen Tsung, Kristie Q. Liu, Jane S. Han, Krutika Deshpande, Tammy Doan, Yong-Hwee Eddie Loh, Li Ding, Wentao Yang, Frank J. Attenello.

**Visualization:** Kathleen Tsung, Kristie Q. Liu, Jane S. Han, Krutika Deshpande, Tammy Doan, Yong-Hwee Eddie Loh, Li Ding, Wentao Yang.

**Writing – original draft:** Kathleen Tsung, Kristie Q. Liu, Jane S. Han, Yong-Hwee Eddie Loh, Li Ding, Wentao Yang, Frank J. Attenello.

**Writing – review & editing:** Kristie Q. Liu, Krutika Deshpande, Josh Neman, Yali Dou, Frank J. Attenello.

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
