## [Decision Letter · Decision Letter 0]

21 Sep 2023

Dear Dr Liu,

Thank you very much for submitting your Research Article entitled 'CRISPRi Screen of Long Non-coding RNAs Identifies LINC03045 Regulating Glioblastoma Invasion' to PLOS Genetics.

The manuscript was fully evaluated at the editorial level and by independent peer reviewers. The reviewers appreciated the attention to an important problem, but raised some substantial concerns about the current manuscript. Based on the reviews, we will not be able to accept this version of the manuscript, but we would be willing to review a much-revised version. We cannot, of course, promise publication at that time.

If you decide to revise the manuscript for further consideration at PLOS Genetics, please aim to resubmit within the next 60 days, unless it will take extra time to address the concerns of the reviewers, in which case we would appreciate an expected resubmission date by email to plosgenetics@plos.org.

We are sorry that we cannot be more positive about your manuscript at this stage. Please do not hesitate to contact us if you have any concerns or questions.

Yours sincerely,

David J. Kwiatkowski

Section Editor

PLOS Genetics

Reviewer's Responses to Questions

**Comments to the Authors:**

Reviewer #1: The authors seek to identify the role of lncRNAs in GBM cell invasion. The long non-coding RNA LINC03045 is identified as a positive hit in a CRISPRi screen using the U87 GBM cell line in an in vitro invasion assay. This result is validated utilizing ASO/CRISPRi KD in vitro with commercially available and patient-derived GBM cell lines. Clinically relevant correlation is shown with TCGA and GTEx analysis demonstrating a relationship between LINC03045 expression and patient outcomes. A further positive association is demonstrated between LINC03045 and WASF3 expression in vitro and in patient cohorts, as well as epistasis through in vitro invasion.

Major Points

1. The study results and conclusion would be most prominently strengthened with some in vivo validation demonstrating either a change in invasion phenotype or animal survival

2. Figure 2. Survival curves should be controlled for grade and IDH status. It is not really relevant to compare lower grade gliomas (which are likely to be IDH mutant) to IDH WT GBMs. In other words, all tumors of a certain grade (you can stick to GBM) should be stratified by LINC03045 expression to determine if expression of this ncRNA is associated with worse prognosis within a specific grade and molecular subtype.

3. Figure 3. I am somewhat surprised by the dramatic in vitro invasion phenotype despite a relatively modest inhibition of RNA expression. Do the authors have any thoughts as to why?

4. A more thorough comparison should be made of the in vivo LINC03045-associated gene expression changes and the in vitro LINC03045 KD-associated changes. Also is here a relationship between WASF3 and any of the GO or KEGG terms that are enriched in the in vivo gene expression analyses?

5. Do the authors know how mechanistically LINC03045 triggers WASF3 expression?

6. Figure 7. Two different siRNAs should be used for WASF3 KD experiments.

7. In epistasis experiments (Fig 8), WASF3 overexpression alone should be presented in the invasion assay.

Minor points

1. As the steps in Fig 8A are not proven in the manuscript, it should be shown in the figure and discussed in the manuscript in a more tempered manner. There is no evidence that STAT3 is involved here and that LINC03045 directly regulates WASF3 expression.

Reviewer #2: In this study, the authors employed the CRISPRi screen method to identify a novel candidate associated with the invasion property in a glioma cell line. They have effectively demonstrated the reproducibility of their invasive assay and have proposed a potential drug candidate for future glioma treatment. The authors conducted CRISPRi screening in the U87 cell line, selected a candidate, and subsequently validated it in three different cell lines, including a patient-derived glioma cell line. However, in order to bolster the reliability of the results, it is imperative to conduct screening experiments in additional cell lines, ensuring that they exhibit validated invasive properties.

Major Comments:

1. To begin with, it is essential to assess the invasiveness of the U87 commercial cell line, potentially through a 3D invasion assay. Given that this cell line may not precisely reflect glioblastoma behavior in real-world scenarios, many researchers prefer employing patient-derived cell lines or tumor spheres. Considering the relatively short duration required for CRISPRi screening, it is advisable to conduct the screening in another cell line, such as USC02 (utilized in the validation experiment), while also demonstrating the invasiveness of this cell line.

2. In Figure 1, the y-axis appears to represent -log P-values, while the x-axis likely represents log fold changes when compared to nontargeting guide RNAs. ZNF215-AS1 emerges as a seemingly more significant candidate. Could the authors provide a rationale for choosing the LINC03045 candidate over ZNF215-AS1? Additionally, in the context of "Positive hits were identified through this process," please specify the positive hits in both the figure and the manuscript. Furthermore, when mentioning, "The dashed lines represent thresholds to determine screen hits," it would be beneficial to elucidate the principle governing the selection of these thresholds. Typically, in cases where the y-axis represents -log P-values, a dashed line at -log(0.05) is used to signify a threshold, with a p-value less than 0.05 indicating a significant finding, as stated in the method section.

Minor Comments:

1. In Figure 1, it is advisable to include a correlation plot illustrating the guide enrichment between the two replicates. This addition would enhance the reliability of the results.

2. Consider converting most of the images into vector format, particularly for figures depicting heatmaps or volcanoplots. This change would improve the clarity and resolution of these figures.

**Have all data underlying the figures and results presented in the manuscript been provided?**

Reviewer #1: Yes

Reviewer #2: **No: **Positive hit in the screenings

PLOS authors have the option to publish the peer review history of their article (what does this mean?). If published, this will include your full peer review and any attached files.

Reviewer #1: No

Reviewer #2: No

---

## [Decision Letter · Decision Letter 1]

21 May 2024

Dear Dr Liu,

We are pleased to inform you that your manuscript entitled "CRISPRi Screen of Long Non-coding RNAs Identifies LINC03045 Regulating Glioblastoma Invasion" has been editorially accepted for publication in PLOS Genetics. Congratulations!

Yours sincerely,

David J. Kwiatkowski

Section Editor

PLOS Genetics

David Kwiatkowski

Section Editor

PLOS Genetics

Comments from the reviewers (if applicable):

Reviewer's Responses to Questions

**Comments to the Authors:**

Reviewer #1: The authors have done a commendable and very thoughtful job of addressing my comments. In vivo would have been nice, but the 3D invasion assay nicely complements their in vitro work with a more physiological platform.

Reviewer #2: Compared to the initial manuscript, the authors have significantly enhanced the quality and presentation of their results. I have a few comments and suggestions for further improvement.

Firstly, I recommend that the authors provide a clean version of the manuscript, as the track-changed format can be difficult for reviewers to read and evaluate.

Secondly, the authors mentioned the challenges of using Cas9-based techniques in patient-derived cell lines. While this can indeed be challenging, I would like to suggest considering lentivirus-based screening methods. In my experience, this approach has been effective with glioma patient-derived cell lines and could potentially be beneficial for their CRISPR-based screening experiments.

Thirdly, the correlation scatter plot in the supplementary figure appears to show a low extent of correlation. It would be helpful if the authors included the correlation coefficient, such as Spearman's or Pearson's, to quantify the correlation and provide a clearer interpretation of the data.

Overall, I believe the manuscript is now suitable for publication.

**Have all data underlying the figures and results presented in the manuscript been provided?**

Reviewer #1: Yes

Reviewer #2: Yes

PLOS authors have the option to publish the peer review history of their article (what does this mean?). If published, this will include your full peer review and any attached files.

Reviewer #1: No

Reviewer #2: No

**Data Deposition**

http://datadryad.org/submit?journalID=pgenetics&manu=PGENETICS-D-23-00531R1

**Press Queries**

---

## [Editor Report · Acceptance letter]

5 Jun 2024

PGENETICS-D-23-00531R1 

CRISPRi Screen of Long Non-coding RNAs Identifies LINC03045 Regulating Glioblastoma Invasion 

Dear Dr Liu, 

We are pleased to inform you that your manuscript entitled "CRISPRi Screen of Long Non-coding RNAs Identifies LINC03045 Regulating Glioblastoma Invasion" has been formally accepted for publication in PLOS Genetics! Your manuscript is now with our production department and you will be notified of the publication date in due course.

With kind regards,

Anita Estes

PLOS Genetics

On behalf of:
